# Chances and Challenges of New Genetic Screening Technologies (NIPT) in Prenatal Medicine from a Clinical Perspective: A Narrative Review

**DOI:** 10.3390/genes12040501

**Published:** 2021-03-29

**Authors:** Ivonne Bedei, Aline Wolter, Axel Weber, Fabrizio Signore, Roland Axt-Fliedner

**Affiliations:** 1Department of Prenatal Medicine and Fetal Therapy, Justus Liebig University Giessen, 35392 Giessen, Germany; aline.wolter@gyn.med.uni-giessen.de (A.W.); roland.axt-fliedner@gyn.med.uni-giessen.de (R.A.-F.); 2Institute of Human Genetics, Justus Liebig University Giessen, 35392 Giessen, Germany; axel.weber@humangenetik.med.uni-giessen.de; 3Department of Obstetrics and Gynecology, Opedale S. Eugenio, 00144 Rome, Italy; fabrizio.signore@asiroma2.it

**Keywords:** non-invasive prenatal testing (NIPT), non-invasive prenatal diagnosis (NIPD), sex chromosome anomaly (SCA), copy number variant (CNV), rare autosomal trisomy (RAT), chromosomal microarray (CMA), whole exome sequencing (WES), whole genome sequencing (WGS)

## Abstract

In 1959, 63 years after the death of John Langdon Down, Jérôme Lejeune discovered trisomy 21 as the genetic reason for Down syndrome. Screening for Down syndrome has been applied since the 1960s by using maternal age as the risk parameter. Since then, several advances have been made. First trimester screening, combining maternal age, maternal serum parameters and ultrasound findings, emerged in the 1990s with a detection rate (DR) of around 90–95% and a false positive rate (FPR) of around 5%, also looking for trisomy 13 and 18. With the development of high-resolution ultrasound, around 50% of fetal anomalies are now detected in the first trimester. Non-invasive prenatal testing (NIPT) for trisomy 21, 13 and 18 is a highly efficient screening method and has been applied as a first-line or a contingent screening approach all over the world since 2012, in some countries without a systematic screening program. Concomitant with the rise in technology, the possibility of screening for other genetic conditions by analysis of cfDNA, such as sex chromosome anomalies (SCAs), rare autosomal anomalies (RATs) and microdeletions and duplications, is offered by different providers to an often not preselected population of pregnant women. Most of the research in the field is done by commercial providers, and some of the tests are on the market without validated data on test performance. This raises difficulties in the counseling process and makes it nearly impossible to obtain informed consent. In parallel with the advent of new screening technologies, an expansion of diagnostic methods has begun to be applied after invasive procedures. The karyotype has been the gold standard for decades. Chromosomal microarrays (CMAs) able to detect deletions and duplications on a submicroscopic level have replaced the conventional karyotyping in many countries. Sequencing methods such as whole exome sequencing (WES) and whole genome sequencing (WGS) tremendously amplify the diagnostic yield in fetuses with ultrasound anomalies.

## 1. Introduction and Overview

The search for prenatal detection of chromosomal anomalies has been ongoing since the 1960s. At the very beginning, maternal age was the screening parameter of choice, and Down syndrome the anomaly that was almost exclusively screened for [1]. Since then things have, at least technically, changed enormously, even though Down syndrome is still the main focus of screening and in the minds of many patients the only diagnosis that they may confront. Structural chromosomal anomalies and monogenic diseases are still not focused on by current screening algorithms, nor by the education system or by society in general [2]. With the clinical implementation of non-invasive prenatal testing (NIPT) in 2012, there has been a paradigm shift in prenatal screening. First trimester combined screening (FTCS) based on maternal age, fetal nuchal translucency thickness (NT) and the serum markers *β*-HCG and PAPP-A has a detection rate (DR) of 90–95%, a false positive rate of 2.5–5% [3] and a PPV of 3.4 [4] for the detection of trisomy 21. Included is the advantage that, if the results are abnormal, it could raise suspicion not only for the common trisomies, but also for early detectable fetal structural defects and other clinically relevant findings such as rare autosomal trisomies (RATs), triploidy, single gene disorders and copy number variants not detectable in a targeted NIPT approach [5]. A disadvantage is the comparatively high false positive rate leading to invasive procedures. The advantage of NIPT for common autosomal trisomies is the high sensitivity and specificity and the very high negative predictive values [6]. In clinical practice this leads to a reduction in invasive testing, now limited to the high risk cases and to confirm NIPT findings [7]. Even though the performance of NIPT is dependent on prevalence i.e., maternal age dependent, overall performance is excellent and superior to first trimester combined screening in the high risk cases and also in the general population, in particular for trisomy 21 [8,9].

Different approaches for the implementation of NIPT have been used. For contingent screening, a group with an elevated risk is defined, who will then be tested by NIPT. Other approaches use NIPT as a first-line screening. However, in some countries no consistent strategy or guidelines exist.

There are concerns about missed diagnosis of atypical chromosomal aberrations in the intermediate and high risk group, when NIPT is offered for common trisomies instead of invasive diagnosis with chromosomal microarray analysis [10]. This knowledge is very important for younger patients and for example egg donation, when the risk for microdeletions or -duplications is higher than that for Down syndrome.

Other considerations are that, by using NIPT as a first-line screening, genetic conditions other than the common trisomies and fetal structural anomalies will be missed or diagnosis could be delayed to the second trimester [11]. There is a consensus that NIPT should always be offered in combination with a qualified ultrasound scan. NIPT can be offered already at around 10 weeks or even earlier. The information of an ultrasound examination at this gestational age is limited to viability and the diagnosis of multiples. Implementation of two ultrasound scans in the first trimester if NIPT is performed early has to be discussed. Offering NIPT at 11+0-13+6 weeks in order to look for increased nuchal translucency and structural fetal anomalies, following the ISUOG guidelines, seems to be the more comprehensive approach [12,13,14]. A recent study showed that management would be changed in almost 10% of cases, when ultrasound was done before blood for NIPT was drawn [15]. Early screening and prevention of pre-eclampsia is also an important factor of first trimester diagnosis and should not be neglected through the exclusive use of NIPT. At the moment it is not clear if the replacement of FTCS for NIPT has an impact on early detection of growth restriction or pre-eclampsia [16].

Besides numeric or structural chromosomal changes, an increased nuchal translucency is also associated with fetal structural malformations (for example cardiac defects, skeletal anomalies, genitourinary tract anomalies and others) and adverse pregnancy outcome [17,18]. Systematic NT measurement can thereby facilitate earlier detection and specialized follow-up [14,19,20].

In several countries, chromosomal microarray has replaced classic karyotyping after invasive diagnosis in general, or in preselected clinical situations such as NT > 99th percentile, structural fetal anomalies, fetal growth restriction < 3rd percentile before 28 weeks, etc. [21,22,23,24]. Chromosomal microarray can detect clinically relevant submicroscopic copy number variants and may change surveillance and decision making in the affected pregnancy [23,24,25]. Furthermore, single nucleotide polymorphism (SNP)-based arrays also allow the detection of triploidy and uniparental disomy [26]. The additional diagnostic yield is dependent on the indication for testing. Wapner et al. found an additional 6% of relevant deletions or duplications in fetuses with normal karyotype and structural anomalies, and 1.7% for the indication “advanced maternal age” or positive screening results [23,27]. Also, diagnostic yield increases with multiple fetal anomalies and is associated with anomalies of special organ systems [28,29].

An elevated risk for pathologic copy number variants is also described for NT > 3 mm in fetuses with normal karyotype [30,31,32,33,34]. This is also the case in altered serum levels of PAPP-A and free *β*-HCG (< 0.2 MoM respectively < 0.45 MoM or > 5 MoM) [11,35].

Another advance has been the inclusion of next-generation sequencing for diagnosis in fetal cells after chorionic villous sampling or amniocentesis [36].

Next-generation sequencing can be applied in different ways:
(1)A clinical exome sequencing (CES) covering genes associated with a known clinical association with disease (Mendeliom) [37].(2)Whole exome sequencing (WES), covering the protein coding segments (exons) of all known genes representing about 1–2% of the genome [38].(3)Whole genome sequencing (WGS) additionally covering regulatory genomic sequences, introns and other non-coding sequences.

With the rapid evolution of genetic testing methods and corresponding bioinformatic capacities, paradigms of screening and diagnostics in prenatal medicine are changing considerably and continuously. Besides counseling issues, ethical issues are also arising and need to be discussed.

Starting as a screening tool for the most common trisomies, cfDNA screening is continuously expanding its spectrum.

Offering the test for sex chromosome anomalies, some providers additionally came up with screening for rare autosomal trisomies, genome-wide deletions and duplications and certain microdeletions, most commonly microdeletion 22q11.2 [39,40,41,42]. In contrast to the convincing evidence for NIPT for trisomies 21, 18 and 13, valid data with respect to accuracy and PPVs are still missing for most of these additional tests [40]. For sex chromosomal anomalies, especially Turner syndrome, the positive predictive value is constantly lower than that for common trisomies and varies between 9% and 40% [43]. Including copy number variants and rare autosomal trisomies with a low incidence and a potentially much lower PPV, rates of invasive testing for false positive results will rise again. In addition, there are concerns about maternal incidental findings and how to proceed and counsel [44]. Thus, screening beyond the common trisomies is currently not recommended by scientific societies until further evidence is available [45,46].

In addition to non-invasive prenatal testing (NIPT) as a screening tool, non-invasive prenatal diagnosis (NIPD) for the detection of single gene mutation, certain monogenic disorders, fetal sex determination and fetal blood group characteristics in pregnancies at risk are used in prenatal medicine [47,48,49,50]. In comparison to aneuploidies and copy number variants, monogenic disorders are so far not reported to have the problem of confined placental mosaicism [51]. Paternally inherited or de novo variants can be diagnosed with more ease compared with autosomal recessive or X-linked disorders. By using dosage-based techniques, variants that are carried by the mother can also be detected noninvasively [51,52]. These approaches are proband-based, mutation-specific and offered to a high risk population. In this setting they are used as a diagnostic tool. Recently NIPD has been picked up by the commercial sector as a screening tool for the general population. One test is offered for de novo mutations in dominant disease genes [53], and another test is offered for recessive diseases including spinal muscular atrophy (SMA), sickle cell disease, thalassemia (α and β) and cystic fibrosis [54]. A combined screening approach has also recently been published including aneuploidies, de novo FGFR3 mutations and paternally derived *β*-thalassemia [55]. It has to be mentioned that these tests are not considered to be diagnostic and thus require confirmation with invasive testing. Data on validation and follow-up are largely missing at the moment.

In addition to the described NIPT, based on fragmented free trophoblast DNA in maternal blood, there are attempts in non-invasive testing for aneuploidies, copy number variants down to 1–2 Mb in size and potentially even monogenic disorders using fetal cells from maternal circulation and cervical mucus [56,57,58,59,60,61]. These approaches provide intact fetal cells i.e., pure fetal DNA without contamination. These techniques are under investigation and not yet applied in clinical practice.

This review will focus on NIPT as a screening tool for sex chromosome anomalies (SCAs), RATs and copy number variants (CNVs).

## 2. cfDNA-Based Screening (NIPT) Other Than Chromosomes 21, 18 and 13

### 2.1. Screening for Sex Chromosomal Anomalies

After the rapid inclusion of NIPT as a reliable screening tool for the common trisomies, screening for sex chromosomal anomalies is now offered by most commercial NIPT providers but was excluded in a genome-wide NIPT program in the Netherlands [62]. Inclusion of sex chromosomal anomalies in a general screening approach has several challenges and should never be offered without extensive pre- and post-test counseling.

Screening for sex chromosomal anomalies includes 45,X (Turner syndrome) and its genetic variants, Klinefelter syndrome (47,XXY), Triple X (47,XXX) and Jacobsen syndrome (47,XYY).

Specific prenatal ultrasound anomalies are uncommon for several sex chromosomal anomalies except Turner syndrome. The spontaneous abortion rate does not seem to be increased in 47,XXY, 47,XXX or 47,XYY fetuses but is considerably high in Turner syndrome. Most Triple X females and Jacobsen syndrome males stay undetected during their lifetime. It is difficult or rather impossible to predict a phenotype for the latter syndrome, and it is even questionable to describe them as pathological conditions per se, which complicates prenatal counseling and decision making. Even though mental retardation is generally not an issue, the rate of induced abortion is still high, especially for fetuses affected by Turner syndrome [63]. An exception is given for karyotypes with more than three X chromosomes (i.e., for example 48,XXXX or 49,XXXXY) when mental retardation becomes considerably prevalent. Current tests are not evaluated for detection of these severe anomalies, and they may “hide behind” a positive NIPT result [64,65].

Detection rates for SCA in a routine first trimester screening for trisomy 21 vary between 42% for Turner syndrome down to 5% for 47,XYY [63].

The PPV for sex chromosomal anomalies, especially for Turner syndrome, is far below the PPV for trisomy 21 and varies between 9% and 40%, also depending on whether ultrasound anomalies are present [43,66].

Different factors contribute to the decreased performance.

Placental mosaicism can be found in 1–2 percent of human placentas [67]. Sex chromosomal anomalies are more prone to feto-placental mosaicism, with monosomy X being the most frequently involved chromosomal anomaly [68].

Other reasons for discordant results may be a vanishing twin, and maternal sex chromosome anomalies, mostly unknown at the moment of NIPT and often not included in pretest counseling [41,69,70].

There is conflicting evidence regarding to what extent age-related X chromosome loss in maternal lymphocytes contributes to a discordant result, and maternal age could be linked with false positive results in some studies, but not in others [43,71].

### 2.2. Screening for Copy Number Variants (“Expanded NIPT”)

Submicroscopic deletions or duplications are structural chromosomal abnormalities that, due to their smaller size, cannot be detected by routine karyotyping with a standard resolution of 5–10 Mb [23,72]. Therefore, they have been missed by conventional cytogenetic testing. With the advent of microarray in prenatal medicine, copy number variants have been shown to be a relevant contributor to prenatal detectable fetal anomalies. Additional diagnostic yield is around 6–9% for fetuses with sonographic detectable anomalies but with a normal conventional karyotype [24]. The detection of microdeletions and -duplications is also increased in fetuses with enlarged nuchal translucency and abnormal serum parameters in first trimester screening. They may be as frequent as 13.8% in fetuses with NT ≥ 3.5 mm [32]. For several pathogenic copy number variants, mental retardation and intellectual disability are common developmental problems as, being a functional abnormality, they are not detectable on prenatal ultrasound [73].

Unlike common trisomies, copy number variants are not linked to maternal age, and statistically younger pregnant women have a higher risk for a submicroscopic aberration than for trisomy 21. This also has to be considered in pregnancies after egg donation.

The risk for a pathogenic copy number variant affecting a pregnancy without any known risk factor or ultrasound anomaly, having a normal karyotype, is between 0.86% and 1.7%. In 0.34% or 1 in 300, this involves an early onset disease [32,73].

Copy number variants are not only linked with genetic disorders but are also important in genetic variation and not necessarily associated with a disease phenotype [72,74,75]. Consistent classification and interpretation can be challenging. Updated guidelines for clinical classification were recently released by the American College of Medical Genetics and Genomics (ACMG) [76].

Thus, copy number variants are more likely classified as pathogenic or likely pathogenic if they involve coding genes, are large in size, and are not found in the parents or normal controls [77].

Microdeletion/-duplication syndromes often show a variable penetrance and expression. They may be inherited by an unaffected or mildly affected parent or alternatively occur de novo. In the absence of ultrasound anomalies, the prediction of a phenotype remains difficult.

In addition to NIPT screening for common trisomies and sex chromosomal anomalies, some companies have added copy number variants to their screening panels. Some have limited their approach to specific microdeletions, for example Di George syndrome (22q11.2), which is the most prevalent microdeletion syndrome with a well-known but also highly variable phenotype [78]. Others report genome-wide approaches detecting gains and losses ≥ 7Mb and losses associated with specific deletions < 7 Mb [39]. Lo et al. reported an algorithm that detects the majority of rearrangements > 6 Mb using a standard next-generation sequencing (NGS) approach not detecting copy number variants < 6 Mb without increasing the sequencing depth [79]. The feasibility of detecting copy number variants at 3 Mb resolution has been demonstrated [80].

There are several factors that affect the performance of NIPT for copy number variants, including the fetal fraction, the size of the copy number variants and the depth of coverage or reads [78,79,81]. With an increasing number of reads, smaller copy number variants can be detected with higher sensitivity but also with increasing costs. The selection of included copy number variants may sometimes be more driven by their size and detection feasibility than by their clinical relevance [82].

The individual incidence of different microdeletions or -duplications is very low and in some cases unknown [78]. This has a tremendous influence on the PPV, which is clearly dependent on the prevalence of a disease. Most information on test metrics is based on smaller studies, smaller patient cohorts but also *in vitro* dilutions of different DNA samples with and without copy number variants [83]. PPVs also vary markedly between a high or low risk population and the applied NIPT technique, making counseling a real challenge [40,83,84,85,86,87].

There are several concerns by professional societies to routinely include screening for copy number variants in NIPT panels [88]. The main aspect is the poor PPV and the expected high rate of false positive results, which in turn have to be validated finally by invasive diagnosis.

On the other hand, it is very difficult to estimate the false negative rate, as confirmatory testing in apparently unaffected individuals is sparse. With a rising number of copy number variants included, or even for a genome-wide approach, obtaining informed consent is nearly impossible in a physician’s daily work due to the complexity of counseling. At the moment, routine screening for copy number variants is not indorsed by professional societies, and further validation studies are needed. If included in prenatal screening, thorough pre- and post-test counseling is essential, as is knowledge of the specific test characteristics of the NIPT used.

### 2.3. Screening for Rare Autosomal Trisomies (RATs)

RATs are trisomies of autosomal chromosomes other than chromosomes 21, 18 and 13. Products of conception (POC) affected by non-mosaic RATs generally do not develop in the early embryonic stages and result in early fetal loss. All RATs have been found in retained products of conception [89]. Karyotyping trophoblast cells from these abortions is possible using chromosomal microarray (CMA) direct preparation, not cultured cells, and typical clusters of chromosomal trisomies are found. cfDNA can also be used to find the underlying reason in cases of miscarriage [90,91].

The majority of RAT cases detected by invasive or noninvasive prenatal testing in viable pregnancies are mosaic trisomies, mostly confined to the placenta (confined placental mosaicism, CPM). The probability that a RAT detected in cytotrophoblast cells represents a true fetal mosaicism is approximately 3% and is lower than for trisomies 21, 18 and 13 and sex chromosomal anomalies [89]. The overall PPV for RATs in NIPT is low or impossible to calculate in different studies, due to the limited numbers [62,86,92]. Other reasons for false positive results not showing the true fetal karyotype may be a vanishing twin. For a diagnostic procedure, amniocentesis is preferred over CVS to avoid influence by confined placental mosaicism, except one that would clear the distribution of cells and describe the mosaicism. This could be important for management in an ongoing pregnancy.

If the fetal karyotype from amniocentesis is normal, there is still a risk to consider of a uniparental disomy of the fetus, which is due to trisomic rescue in early stages after conception resulting in a mosaicism, not diagnosed in cells from amniotic fluid after correction of the trisomy. Such findings are reported mostly if the chromosomes 6, 7, 11, 14, 15 and 20 have been involved, but not all are of clinical importance [67,93].

Outcomes with an unremarkable course of pregnancy have been reported, linked to the chromosome involved, with particularly poor outcome for trisomies 15, 16 and 22, and a mostly favorable outcome for trisomy 7, the most frequent RAT found by NIPT [89,94,95]. Trisomy 16 is associated with severe intrauterine growth restriction (IUGR), pre-eclampsia and fetal anomalies. The risk of an adverse outcome in a NIPT positive for trisomy of chromosome 16 is about 64.5% [89,96]. Because most of the RATs are very rare and cases reported are most often in mosaic state with variable tissue involvement, an exact prediction of their effects for the particular pregnancy is difficult [89]. The complexity of the genetics of RATs and the required testing strategy (NIPT and invasive testing) requires intense counseling to avoid unnecessary stress and anxiety for the parents. However, findings may guide us towards a more intense pregnancy surveillance for early detection of IUGR and stillbirth prevention. More studies are needed to better answer these questions, and professional societies do not recommend RATs in routine NIPT screening.

Multiple aneuploidies/complex abnormal NIPT results have been described in cases of maternal malignancy occurring during the current pregnancy. This phenomenon as an accidental finding is not only linked to rare autosomal trisomies but may also include multiple common chromosomal anomalies or fragmented DNA gains and losses [97]. How to proceed in these cases is difficult, depending on pretest counseling and patient information after an unexpected result [98]. In the TRIDENT-2 trial, in 81% of pregnant patients with complex NIPT profiles, a malignancy or maternal leiomyomas were found [62]. Thus, for pregnant women with known malignancy, NIPT is not recommended.

## 3. New Diagnostic Tools in Prenatal Diagnosis

For decades the karyotype has been the gold standard in prenatal medicine. The presentation of mitotic chromosomes at a resolution between 5 and 10 Mb is appropriate for aneuploidies, large chromosomal rearrangements and higher grade polyploidies. Microdeletions and -duplications, single gene mutations and UPD or other epigenetic disorders cannot be detected. Karyotyping is labor-intensive and limited to vital cells to be grown in culture. In general, it takes between one and two weeks after biopsy to get the results of a long-term culture.

### 3.1. Chromosomal Microarray Analysis

Genetic anomalies are prenatally found with an a priori prevalence of 3%, with aneuploidies, microdeletions/microduplications (including CNVs) and single gene disorders almost equally distributed, i.e., 1 in 100 cases for each entity [37]. Copy number variants and monogenic diseases require special techniques of molecular cytogenetics and molecular sequencing (Sanger sequencing and next-generation sequencing) and can clearly not be detected by karyotyping. Microdeletions as a loss of genetic information are responsible for significant anomalies and developmental delay and associated with severe functional disease, for example epilepsy. Duplications are also the underlying cause in developmental and dysfunctional disabilities.

In rare situations it remains difficult—despite well curated data collections and tools based on artificial intelligence (AI)—to predict an individual phenotype in the prenatal setting. These problems can often be solved by trio analysis: the index patient and the parents.

Technically, there are two different types of microarrays: SNP arrays, which are based on single nucleotide polymorphism, and array CGH (aCGH) based on comparative genomic hybridization, to detect derivations in copy numbers relative to a control genome. The allelic information from the single nucleotide polymorphism enables the description of polyploidies and uniparental disomy.

Chromosomal microarray analysis was introduced in the prenatal setting in 2005 and could demonstrate an additional diagnostic yield in 5/49 (about 10%) fetuses with multiple anomalies [99]. It was shown in recent following studies that inclusion of microarray analysis in prenatal diagnostics for different indications leads to an addition rate of pathogenic findings between 1.7% and 9% if the karyotype is normal [23,24,31].

Depending on the selection criteria, these numbers can be even higher. The highest diagnostic yield can be found in fetuses with multiple anomalies, cardiac and renal anomalies [28].

Balanced chromosomal anomalies cannot be detected by microarrays if they are proven to be truly balanced, i.e., no gain or loss can be found at the translocation breakpoints previously diagnosed by karyotyping. The probability that prenatally an abnormal fetal phenotype is associated with an unbalanced translocation due to the location of breakpoints and a putative disruption of one or more genes is estimated to be around 6% [100]. CMA can be indicated in so-called “balanced” translocations to reassure the complete DNA structure.

Microarrays can also help in delineating marker chromosomes [100].

Detection of copy number variants by chromosomal microarray analysis is directly dependent on the array platform/chip used. Array platforms differ in the coverage of affected genomic regions and thus the detectable size of copy number variants within these regions [72,101].

Like sequence variants, copy number variants can also be classified as variants of uncertain significance, complicating genetic counseling. With the increasing knowledge of copy number variants classification, this phenomenon is decreasing over time, and it is currently 1–2%, or even lower [27,101]. Analysis of the parental blood can also help in classification.

The detection of late-manifesting diseases by chance, difficulties in counseling in diseases with variable penetrance, and incidental findings in the fetus and the parents are also challenging ethical problems [102].

In many countries, chromosomal microarray analysis has replaced conventional karyotyping and is used as a first-line diagnostic technique, mandatory in special indications. Classical karyotype is added to answer specific questions, for example to differentiate between free trisomies or those by unbalanced Robertsonian translocations.

### 3.2. Genome-Wide Sequencing

New sequencing methods (next-generation sequencing, NGS) allow the investigation of multiple genes of a fetus in parallel in a reasonable time, which is crucial in prenatal diagnosis. There are several approaches that can be applied. One is to test not only for a single gene, but for multiple genes that could be linked to a specific or overlapping phenotype (targeted gene panels). For unspecific phenotypes, genome-wide approaches are more feasible. It is possible to include all known disease genes in a clinical exome (CES), all known genes including those not yet linked to a specific phenotype or disease (whole exome sequencing, WES), or to cover the sequence of the whole genome (WGS), including regulatory, non-coding and mitochondrial DNA [103].

Mostly clinical exome and whole exome sequencing, representing about 1–2% of total genomic DNA, are applied in clinical practice. If parental samples are directly tested in addition to the fetus (trio exome), bioinformatical analysis and interpretation of sequence variants can be optimized.

Evidence shows that there is an increase in diagnostic yield using next-generation sequencing approaches in a sequential way after conventional techniques such as karyotype and/or chromosomal microarray analysis in prenatal medicine. The amount of additional diagnostic information varies largely with the indication for testing [37,104]. The range of additional diagnostic yield in fetuses with structural anomalies that could be found in the literature varies considerably, depending on different factors [105]. In the PAGE study, diagnostic yield over conventional testing was highest in fetuses with multiple anomalies (15.4%), skeletal anomalies and cardiac defects. Isolated increased NT in the first trimester had the lowest yield (3.2%) in cases with normal karyotype and normal chromosomal microarray analysis [106]. In practical terms the target population for next-generation sequencing at the moment, if not performed as a first-line approach, includes fetuses with structural anomalies and increased NT.

It is important to understand that, even combining karyotyping, chromosomal microarray analysis and next-generation sequencing approaches, there is no guarantee to find a causal genetic variant in all cases. Therefore, it is important to include in counseling both the various aspects of the procedure and informed consent for the clinical suspicion.

Variant analysis and interpretation is still a challenge per se and compared with the postnatal setting, prenatally it is often restricted to including an often unspecific or incomplete fetal phenotype in the interpretation. Depending on the gestational age, the phenotype may still be incomplete or at least incompletely detectable, because anomalies may arise sequentially or late in pregnancy [107]. Furthermore, for many postnatally well described diseases, we do not know the prenatal phenotype in detail. Lethal variants may be missed because they are under-represented in postnatal databases. Currently a database based on anomalies found in prenatal ultrasound with a correlation of fetal phenotypes and genotypes does not exist. Another issue is the question of the reporting or disclosure of incidental findings and genetic variants for late-manifesting diseases, in the fetus, but also in the parents, in the case of next-generation sequencing trio approaches. The joint position statement from the International Society for Prenatal Diagnosis (ISPD), the Society for Maternal Fetal Medicine (SMFM) and the Perinatal Quality Foundation (PQF) gives detailed advice on what to include in pre- and post-test counseling [108].

## 4. Conclusions

Several new diagnostic tools have changed and challenged the modus operandi in prenatal diagnosis in recent years. The following points are important for the prenatal medicine specialist and all doctors dealing with counseling of pregnant women.
NIPT is not a diagnosis. NIPT for the common trisomies, mostly trisomy 21, has the best performance of all screening approaches. Performance for trisomy 13 and 18 is still very good, more comparable to combined first trimester screening, including early anomaly scan [37]. This evidence is for singleton but also twin pregnancies.NIPT performance for sex chromosomal anomalies, mostly monosomy X, is much worse, due to biological factors. This should be taken into consideration in genetic counseling. Expansion of NIPT beyond the common trisomies is promising, but raises several technical and ethical challenges, which should be addressed.NIPT should always be combined with a skilled ultrasound examination.Invasive prenatal testing provides diagnosis. In invasive prenatal diagnostics, the use of chromosomal microarray analysis, and more recently, next-generation sequencing approaches, has expanded the prenatal diagnostic yield considerably. Next-generation sequencing approaches should be used additionally in fetal anomalies with a normal routine testing result (karyotype and/or chromosomal microarray analysis), not as a first-line test. Using these new techniques, parental pre- and post-test counseling is mandatory, and close collaboration in a multidisciplinary team is urgently needed.Adequate genetic pre- and post-test counseling is mandatory to ensure an informed consent of the patients as the basis for all genetic testing strategies and furthermore, to respect the right not to know if desired.

## Data Availability

Not applicable.

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
