# Peer review of "Chances and Challenges of New Genetic Screening Technologies (NIPT) in Prenatal Medicine from a Clinical Perspective: A Narrative Review"

_genes, 2021, doi:10.3390/genes12040501_

Round 1
Reviewer 1 Report
The review "Chances and challenges of new genetic screening technologies (NIPT) in prenatal medicine from a clinical perspective - a narrative review" is well performed, it represents a good work for all the obstetricians and clinicians who would like to have more detailed informations on new genetic screening technologies. The English language is appropriate. I suggest to better clarify the aspect of NGS: the authors assume that "NGS approaches
should be used additionally in fetal anomalies with a normal routine testing result (karyotype and/or CMA), not as a first-line test", so which is the target population of NGS, considering that they say "it is important to understand that, even combining karyotyping, CMA and NGS approaches, there is no guarantee to find a causal genetic variant in all cases.". Another aspect which needs to be clarified is related to counselling: what is the proposal of the authors? to have a specific counselling related to the procedure (karyotyping, CMA and NGS approaches), or a specific counselling and informed consent related to the clinical or ultrasonographic suspicion? At the end of the review I suggest to the authors to give clearer messages , which could led the clinical activity in a better way and could increase the readers' interest
Author Response
Dear Reviewer,
thank you very much for the important remarks.
- to better clarify the aspect of NGS: We thank you for this important point and inserted, that at the moment the target population for NGS should be in practical terms fetuses either with structural anomalies or isolated increased NT after conventional testing or CMA evaluation.
- Another aspect which needs to be clarified is the the question of counselling.. Thank you for this important point. We included the sentence: It is important to encounter into counselling both, aspects of the procedure and informed consent to the clinical suspicion.
- At the end of the review..Thank you for this interesting point. We reorganized the conclusion section accordingly.
Reviewer 2 Report
Thank you for a nice article. Why did you decide not to include NIPD and carrier screening in the article?
Please correct a few mistakes and explain all the abbreviations.

Author Response
Dear Reviewer, thank you for your important remarks.
Why did you decide not to include..: Within the section 2.2 Screening for CNVs (expanded NIPT) there is already a section dealing with NIPD and carrier screening beginning with: In addition to non-invasive prenatal testing as a screening tool...
Pls explain all abbreviations: We thank you for this important point and included a list of abbreviations at the end of the text
Reviewer 3 Report
Comments to genes-1143839 entitled
Chances and challenges of new genetic screening technologies (NIPT) in prenatal medicine from a clinical perspective - a narrative review
The current study attempted to review the recent chances and challenges of new genetic screening technologies (NIPT) in prenatal medicine. In general, it is very difficult for me to read this article.
To many abbreviations and abbreviations with expanding words repeated again and again. Relatively confusing to audience, at least for me.
It is also confusing that the authors reported “FTCS” in the section.
Sorry to say that I cannot follow the order of the current article. It is hard to me to review this article. Extensive revision had better be suggested.
It need table to show the advantages of NIPT for the first line in prenatal screening, compared to other methods. The authors also show the blood test or sonography examination.
Author Response
Dear Reviewer, thank you for your important comments.
- FCTS: First trimester combined screening was the basis of prenatal non invasive testing the last 25 years, therefore we mentioned it in one sentence at the beginning of the introduction.
- Too many abrreviations: We tried to reduce the abrreviations throughout the article.
Round 2
Reviewer 3 Report
present version manuscript's has improved, don't have other question